# Global Projections of Heat-Stress at High Temporal Resolution Using Machine Learning

Pantelis Georgiades[1,2], Theo Economou[2,4], Yiannis Proestos[2], Jose Araya[2], Jos Lelieveld[2,3], and Marco Neira[2]

[1]Computation-based Science and Technology Research Center CaSToRC, The Cyprus Institute, Nicosia, Cyprus
[2]Climate and Athmosphere Research Centre (CARE-C), The Cyprus Institute, Nicosia, Cyprus
[3]Max Planck Institute for Chemistry, Mainz, Germany
[4]Department of Mathematics and Statistics, University of Exeter, Exeter, UK

**Correspondence:** Pantelis Georgiades (p.georgiades@cyi.ac.cy) and Marco Neira (m.neira@cyi.ac.cy)

**Abstract.**

Climate change poses a significant threat to agriculture, with potential impacts on food security, economic stability, and human livelihoods. Dairy cattle, a crucial component of the livestock sector, are particularly vulnerable to heat stress, which can adversely affect milk production, immune function, feed intake, and in extreme cases, lead to mortality. The Temperature Humidity Index (THI) is a widely used metric to quantify the combined effects of temperature and humidity on cattle. However, the THI was previously estimated using daily-level data, which does not capture the daily thermal load and cumulative heat stress, especially during nights when cooling is inadequate. To address this limitation, we developed a machine learning approach to temporally downscale daily climate data to hourly THI values. Utilizing historical ERA5 reanalysis data, we trained an XGBoost model and generated hourly THI datasets for 12 NEX-GDDP-CMIP6 climate models under two emission scenarios (SSP2-4.5 and SSP5-8.5) extending to the end of the century. This high-resolution THI data provides an accurate quantification of heat stress in dairy cattle, enabling improved predictions and management strategies to mitigate the impacts of climate change on this vital agricultural sector.

## 1 Introduction

Climate change, driven by anthropogenic greenhouse gas emissions, is a multifaceted challenge with profound implications for ecosystems and human societies alike (IPCC, 2023; Malhi et al., 2020). The agricultural sector, which has been the cornerstone of global food security and economic activities for the past centuries, is particularly vulnerable to climate change and variability (Abbass et al., 2022). Within this sector, livestock farming emerges as a critical area of concern due to its susceptibility to environmental stressors, making the assessment and management of climate impacts critical for sustaining agricultural productivity and livelihoods (Cheng et al., 2022; Escarcha et al., 2018).

Dairy farming, an integral component of the livestock industry, is particularly sensitive to climatic conditions. Economic losses due to heat stress in the US alone are estimated at \$1.5 - \$1.7 billion per year, accounting for approximately 63.9% of

the national yearly losses of this economic sector (North et al., 2023; St-Pierre et al., 2003; Cartwright et al., 2023). Predictive models for the US forecast monetary losses as high as $2.2 billion by the end of the century (Mauger et al., 2014) .

The effects of heat stress on cattle are determined by complex interactions between environmental factors (particularly temperature and humidity) and biological parameters. Modern-day breeds of dairy cattle are the result of intensive genetic selection, aimed primarily at increasing milk productivity. However, this increased productivity is genetically linked to physiological traits such as greater metabolic rates and increased feed intake, both of which augment endogenous heat generation in the animals, thereby making high-productivity breeds particularly susceptible to heat stress (Kadzere et al., 2002; Moore et al., 2023a).

Dairy cows depend on evaporative heat loss as their main thermoregulatory mechanism (Zhou et al., 2023). Therefore, when exposed to increased temperatures, they rely heavily on their ability to dissipate heat by either sweating or panting in order to regulate their body temperature. Water evaporation rates are negatively correlated with the relative humidity of the surrounding environment, so a cow's ability to regulate its body temperature is progressively diminished with increasing moisture in the air (Bohmanova et al., 2007). As a consequence, even moderate increases in temperature can have severe biological repercussions under high-humidity conditions. Heat stress has been linked to multiple deleterious effects in dairy cattle, including reductions in milk yield and quality, decreased reproductive success, decreased feed intake, body-weight loss, reduced immune function, altered behaviour and, in extreme cases, mortality (Burhans et al., 2022; Cartwright et al., 2023; Kadzere et al., 2002; Polsky and von Keyserlingk, 2017).

The Temperature Humidity Index (THI) is a robust, non-invasive metric developed to quantify the levels of thermal stress caused by the combined effects of temperature and humidity on cattle. Its calculation requires meteorological data that is generally easy to access (i.e. air temperature and relative humidity), and its correlation with physiological parameters has been validated by a large body of literature, (Bohmanova et al., 2007; Ravagnolo et al., 2000; Bouraoui et al., 2002; Brügemann et al., 2012; Igono et al., 1992; Bernabucci et al., 2014). For example, THI values above 68 have been associated with reductions in milk yield in dairy cows (Moore et al., 2023b; Collier et al., 2012; Zimbelman et al., 2009), and a recent systematic review of the scientific literature published during the last two decades about the effects of THI on dairy cattle, found that values above 68.8 were associated with increased mortality and reduced fertility, in addition to reductions in milk production (North et al., 2023). The THI can also be used for the definition and classification of heatwaves in relation to their effect on cattle (Hahn et al., 1999, 2009), and it constitutes the basic metric for the Livestock Weather Safety Index, an early warning system which provides specific THI thresholds for normal (THI $\leq$74), alert (THI 75-78), danger (THI 79-83), and emergency (THI $\geq$84) climatic conditions (Hahn et al., 2009).

In most of the available scientific literature, THI values are estimated using daily-level data (e.g. daily averages or daily extremes in temperature and humidity, etc.). The reason for this is twofold: On one hand, working at finer temporal resolutions (e.g. hourly) generally requires the processing and storage of very large datasets, which can pose logistic and computational difficulties. On the other hand, data provided by climate projections of future scenarios is only available at daily or coarser temporal resolutions. Unfortunately, daily-level calculations can neither accurately estimate the daily thermal load caused by fluctuating climatic conditions across each day (e.g. diurnal vs nocturnal temperatures), nor capture cumulative effects over

consecutive days, particularly during periods such as heatwaves, when night-time conditions might not allow for efficient heat dissipation (St-Pierre et al., 2003; Hahn, 1997; Hahn et al., 2009). This underscores the need for increasing the temporal resolution of climate projections in order to reflect the environmental stressors impacting dairy cattle, thereby allowing for

improved forecasts of the potential impacts of climate change on this key economic sector.

Recent decades have seen significant advances in computational capabilities, allowing machine learning algorithms to improve the spatial and temporal resolution of climate data (Huntingford et al., 2019). These innovations enable the downscaling of global climate model outputs to produce high-resolution projections that better address the needs of agricultural planning and management. However, despite progress in spatial downscaling through traditional statistical methods (Nyeko-Ogiramoi

et al., 2012; Tang et al., 2016) and artificial intelligence techniques (Rampal et al., 2022; Pour et al., 2016; Ashiotis et al., 2023), studies focused on temporal downscaling remain scarce. Most recent research has primarily concentrated on downscaling precipitation data with restricted spatial coverage, predominantly employing traditional statistical approaches rather than machine learning methodologies (Requena et al., 2021; Michel et al., 2021). A notable exception is the work by Wang et al. (2024), who demonstrated the capability of deep learning models to temporally downscale temperature data, albeit at a regional

level.

Traditionally, two methodologies have been employed for temporal downscaling of climatic data: dynamical and statistical. Dynamical downscaling involves physical models but is often prohibitively expensive in terms of computational resources for long-term, global applications that require relatively high spatial resolution. In contrast, statistical methods are data-driven and focus on extrapolation using auxiliary parameters. Machine learning, as an advanced form of statistical downscaling, leverages

large datasets to capture complex patterns and dependencies. Our analysis aims to provide improved estimates of both the duration and intensity of heat stress periods for cattle on an hourly basis, integrating data on expected diel fluctuations in Temperature-Humidity Index (THI) values, using a highly scalable machine learning approach that accommodates multi-year, multi-model, and multi-scenario analyses. This need stems from the fact previous work relied on daily-level data, which only allow for approximate estimations of these fluctuations through simplified mathematical models. For instance, St-Pierre et al.

(2003) modeled the intensity of heat stress in the United States by assuming a perfect counter-cyclical relationship between temperature and humidity, with THI variations following an ideal sine wave pattern. While such idealized models can be useful in the absence of high-resolution temporal data, they often overlook the inherent complexities of climatic cycles, such as the influence of geographic diversity and seasonal variations.

Our study aims to bridge the gap between coarse-resolution climate projections and the fine-scale environmental data re-

quired for effective farm management under changing climatic conditions.

## 2 Methodology

We utilized a well-established machine learning algorithm, specifically the 'Extreme Gradient Boost' (XGBoost) model, to temporally downscale daily climate projections to hourly THI values. We opted for the XGBoost model for its computational efficiency compared to Random Forest and other analogous algorithms, specifically for our application. Additionally, the

implementation of Random Forest in Python does not support incremental learning, which was crucial for this study due to the vast amount of data the model needed to process during training. Furthermore, the model was trained on CPU rather than GPU due to memory limitations of our available GPUs and the extensive nature of our dataset.

Our approach involves the training of the model using the ERA5 reanalysis dataset, which contains historical hourly data (Hersbach et al., 2020). The model was subsequently applied to generate hourly THI projections until the end of the century,

based on bias-adjusted climate projections from the NASA NEX-GDDP-CMIP6 datasets (Thrasher et al., 2022). We developed data using twelve climate models and concentrated on two distinct Shared Socioeconomic Pathways (SSPs): SSP2-4.5 and SSP5-8.5, which represent moderate and high greenhouse gas emissions scenarios, respectively, aiming to capture a broad range of potential climatic outcomes.

## 2.1   Data

Two distinct sources for climate data were used in this study: ERA5 reanalysis and NEX-GDDP-CMIP6. Details on each one are provided below.

### 2.1.1   ERA5 reanalysis

The ERA5 reanalysis data, produced by the European Centre for Medium-Range Weather Forecasts (ECMWF) as part of the Copernicus Climate Change Service, combine historical observations into global estimates using forecasting models (Hersbach

et al., 2020). This data set is provided at a spatial resolution of $0.25°$ and hourly temporal resolution (atmosphere component). For the purposes of this study, we retrieved the variables *t2m* (temperature at 2 m) and *d2m* (dewpoint temperature at 2 m), for a time period spanning from 1980 to 2020, from the "ERA5 hourly data on single levels from 1940 to present" entry available in the Copernicus Data Store (CDS), using the Python API.

We estimated the relative humidity variable using the Magnus formula (WMO, 2021), as follows:

$$e(T_d) = 6.1078 \cdot \exp\left(\frac{17.1 \cdot T_d}{235 + T_d}\right) [hPa] \tag{1}$$

$$e_s(T) = 6.1078 \cdot \exp\left(\frac{17.1 \cdot T}{235 + T}\right) [hPa]. \tag{2}$$

where T and $T_d$ are the ambient and dew point temperature in degrees Celsius, respectively. $e(T_d)$ is the vapor pressure at temperature $T_d$ and $e_s(T)$ the saturation vapor pressure at temperature $T$. Finally, the relative humidity can be calculated by taking the ratio of the two, as follows:

$$RH = 100 \cdot \frac{e}{e_s} [\%] \tag{3}$$

The ground truth THI values were derived from the ERA5 reanalysis dataset, as detailed in Section 2.2. This dataset represents the current state-of-the-art for global atmospheric condition proxies, integrating sophisticated numerical model simulations with assimilated observational data. Its performance has been validated in the scientific literature (Bell et al., 2021; Tarek et al., 2020). Furthermore, Napoli (2020) demonstrated its capacity for estimating thermal stress and discomfort indices. Moreover, ERA5 offers a continuous global time-series, which was crucial to our study.

### 2.1.2 NEX-GDDP-CMIP6

The NEX-GDDP-CMIP6 ensemble dataset comprises global downscaled climate change scenarios. These were derived from the General Circulation Model (GCM) runs conducted under the Coupled Model Intercomparison Project Phase 6 (CMIP6) (Thrasher et al., 2022). It includes global downscaled and bias adjusted projections from ScenarioMIP model runs, and features a 0.25° spatial resolution and daily temporal resolution. The data for twelve climate models and two greenhouse gas emissions scenarios (SSP2-4.5 and SSP5-8.5) were retrieved in netCDF format from the NCCS THREDDS data service. From these datasets, we utilized the daily average, minimum and maximum temperatures, as well as the mean relative humidity variables. Table 1 presents the full list of NEX-GDDP-CMIP6 models used in this study to generate hourly THI projections until the end of the century.

**Table 1.** List of NEX-GDDP-CMIP6 models used in this study to generate hourly THI predictions.

| No. | Model Name | No. | Model Name |
|-----|------------|-----|------------|
| 1 | ACCESS-ESM1-5 | 7 | GFDL-ESM4 |
| 2 | CMCC-CM2-SR5 | 8 | INM-CM4-8 |
| 3 | EC-Earth3 | 9 | INM-CM5-0 |
| 4 | EC-Earth3-Veg-LR | 10 | MIROC6 |
| 5 | FGOALS-g3 | 11 | MRI-ESM2-0 |
| 6 | GFDL-CM4 | 12 | NorESM2-MM |

## 2.2  Feature selection

The Temperature Humidity Index is not a directly measured physical quantity, but rather a calculated metric derived from temperature and relative humidity (Cheng et al., 2022). In this study, we used the ERA5 reanalysis dataset to compute THI values, which was the target variable for our machine learning models. THI values were calculated using a computational approach that preserved the spatial and temporal resolution of the original ERA5 data, 0.25° and hourly, respectively.

The computation of hourly THI values from the ERA5 dataset was performed using the following formula:

$$THI = (1.8 \times T + 32) - (0.55 - 0.0055 \times RH) \times (1.8 \times T - 26) \tag{4}$$

where $T$ denotes the temperature in degrees Celsius ($°C$) and $RH$ represents the relative humidity in percentage (%) (Yeck, 1971). This approach ensures that our derived THI values are systematically calculated across the entire spatial and temporal domain of the ERA5 dataset, providing a consistent and comprehensive representation of thermal comfort conditions.

To ensure compatibility with the variables available in the NEX-GDDP-CMIP6 datasets, we generated features from the hourly ERA5 dataset as follows:

- Daily minimum, maximum and average temperature.

- Daily average THI, calculated using the daily average temperature and average relative humidity.

- Daily average relative humidity

Lastly, we included the 'hour of the day' and 'day of the year' features to account for diurnal and seasonal variations of THI, and the land-sea mask -ranging from 0 (sea) to 1 (land)- to differentiate between terrestrial and maritime environments.

## 2.3 Data workflow

This section outlines the utilization of ERA5 reanalysis data and CMIP6 projections in constructing the input variables for this study. Figure 1 provides a high-level overview of the data pipeline procedures employed to train and implement a machine 150 learning model that temporally downscales daily data to hourly THI values.

  To allow for a one-to-one relationship between the hourly ERA5 and daily CMIP6 data, daily features were constructed from ERA5, which are also available in the projection datasets; namely daily average relative humidity and temperature, and daily maximum and minimum temperature. For each day the daily averaged relative humidity and temperature were used to calculate the daily averaged THI. These features were used with no modification from the CMIP6 dataset.

Subsequently, to build the training set, we calculated two additional features, based on the location of each grid cell (lon, lat) and the date; namely the length of the day (number of hours for each grid cell that experienced sunshine for each day), and the day of the year (1-366 to account for leap-years). The day length was calculated using the Brock model (Brock, 1981). In this model, the day length is defined at the point where the centre of the sun is even with the horizon. The declination of the Earth is calculated by (Forsythe et al., 1995):

$$\phi = 23.45 * sin(\frac{283 + J}{265}) \tag{5}$$

where J is the day of the year. The sunrise/sunset hour-angle is calculated as:

$$hourAngle = cos^{-1}(-tan(L)tan(\phi)) \tag{6}$$

where L is the latitude. Finally, day length ($D$) is calculated by:

$$D = 2 * \frac{hourAngle}{15}. \tag{7}$$

The hourly THI value, calculated from hourly relative humidity and temperature, was used as the target variable for the model during training (depicted in red in Fig 1).

To establish a one-to-one relationship between the hourly ERA5 data and the daily CMIP6 data, daily features were constructed from the ERA5 dataset that are also available in the projection datasets. These features include daily average relative humidity, daily average temperature, and daily maximum and minimum temperatures. For each day, the daily averaged relative

humidity and temperature were used to calculate the average THI. These features were used without modification from the CMIP6 dataset, as they are available on a daily temporal resolution already. To construct the training set, we calculated the two additional features, day length and day of the year.

Furthermore, these daily values were combined with a land-sea mask for each grid cell to account for the distinction between coastal and land-locked grid cells. An additional feature, the hour of the day (ranging from 0 to 23), was also included to create

the hourly training set. The resulting hourly dataset was utilized to train the model for predicting hourly THI, with the hourly THI serving as the target variable for this analysis.

Similarly, for inference, the daily CMIP6 data were combined with features representing day length, day of the year, land-sea mask, and hour of the day to construct the hourly datasets utilized in the inference procedures. Table 2 shows the features used for each time-step and their respective temporal resolution.

**Table 2.** Feature set and temporal resolution of each feature. This represents the input variables used in each time step (hourly) of the model to temporally downscale daily data to hourly THI values.

| Feature name | Short name | Temporal resolution |
|---|---|---|
| Average THI | THI_ | Daily |
| Average Relative Humidity | rhmean | Daily |
| Average Temperature | t2m | Daily |
| Minimum Temperature | t2min | Daily |
| Maximum Temperature | t2mmax | Daily |
| Day Length | dayLength | Daily |
| Day of the Year | dayOfYear | Daily |
| Land Sea Mask | lsm | Constant |
| Hour of the Day | hourOfDay | Hourly |

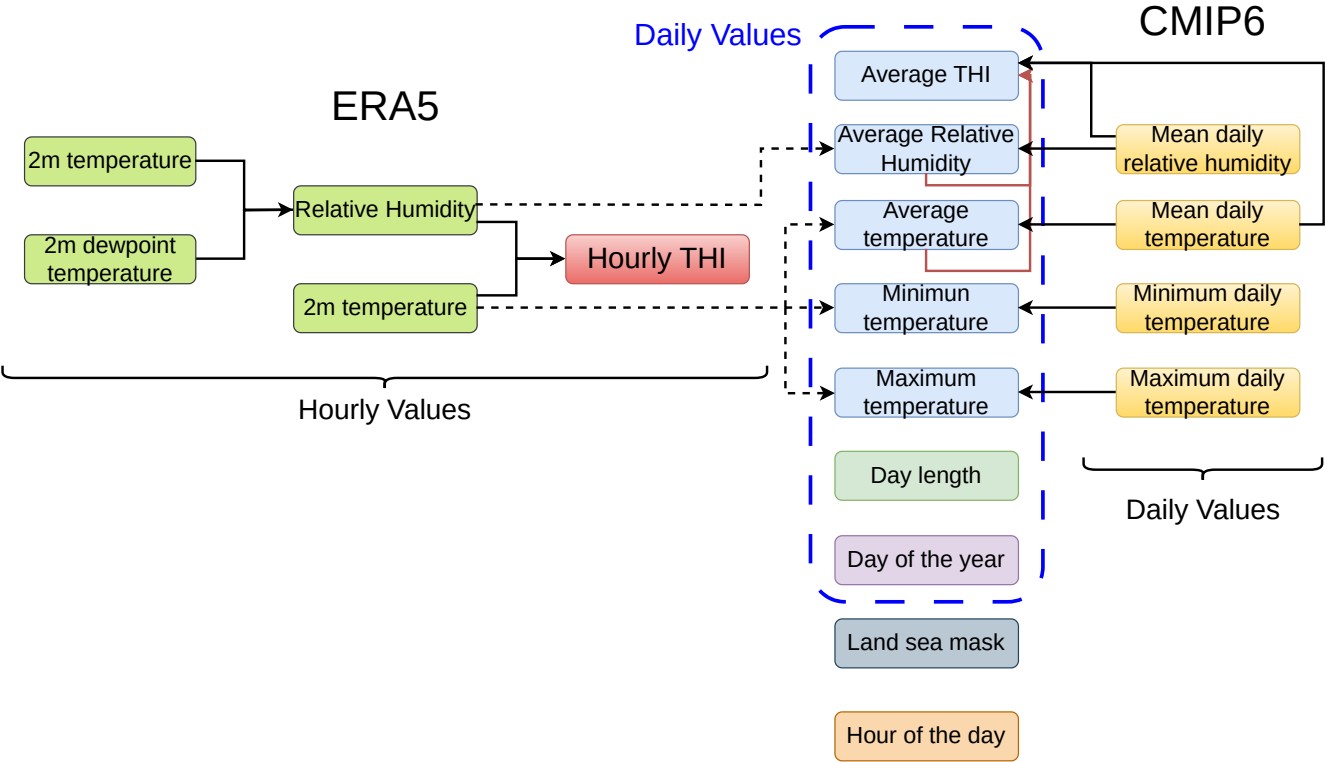

**Figure 1.** High level overview of the data work-flow employed in this study to temporally downscale daily data to hourly THI values. On the left, the data originating from the ERA5 data is presented, whereas on the right, the CMIP6 data is presented. The column in the middle represents the feature set employed in the study to train and perform inference procedures.

## 2.4 Model Training

An XGBoost regressor model was employed to perform the temporal downscaling from daily to hourly resolution. Three models of increasing complexity were trained to explore the trade-off between model performance and computational efficiency. The parameters of each of these models is presented in Table 3.

**Table 3.** The parameters of the three XGBoost models trained to temporally downscale daily climate data to hourly THI values.

| Model no. | Lambda regularisation | Max depth | Number of parallel trees | Learning rate |
|---|---|---|---|---|
| 1 | 1 | 5 | 10 | 0.1 |
| 2 | 1 | 5 | 20 | 0.1 |
| 3 | 5 | 6 | 30 | 0.01 |

The selection of the predictive model was partially influenced by the necessity for an approach capable of incremental learning. This requirement was dictated by the sheer volume of the training dataset, which precluded the possibility of training on the entire dataset simultaneously, due to technical limitations. The *xgboost* library, implemented in Python, was chosen for its ability to accommodate this need as well as the well-established accuracy and speed compared to other ensemble learning models (Chen and Guestrin, 2016; Sheik et al., 2024). The framework facilitated the training of the model in monthly increments, commencing from the year 1980 and concluding in 2017. To ensure the continuity and assess the model's performance over time, checkpoints were stored at the end of each training increment (monthly). The first month of 2018 was used as a test set throughout the training procedures. Finally, the models were trained on a single compute node, which was equipped with two AMD EPYC/Milan 64-core CPUs and 256 GB of RAM. During both the training and inference phases, each model was configured to utilize 128 parallel processes, optimizing computational efficiency. In total, the models were trained on approximately 130 billion examples; areas comprised entirely of sea or ocean were omitted.

## 2.5 Model Evaluation

The performance of the trained models was assessed using ground truth data derived from the ERA5 dataset for the period spanning from February 2018 until December 2020, which wasn't seen by the model during training. This evaluation phase aimed to establish the models' predictive accuracy and their ability to generalize to unseen data. Model performance was quantitatively evaluated using standard statistical metrics, including the Mean Error (ME), Mean Squared Error (MSE), Mean Absolute Error (MAE) and coefficient of determination ($R^2$).

### 2.5.1 Implementation Details

The data manipulation and feature engineering were performed using Python 3.11, utilizing the *xarray*, *numpy*, and *pandas* libraries. The input variables were scaled to the 0-1 range using the *MinMaxScaler* method from the *scikit-learn* library, and the *xgboost* library was used to implement training and inference procedures for temporally downscaling daily climatic variables to hourly THI values.

For the XGBoost regression model, we used the *xgb.Booster()* method, with each training epoch –corresponding to a month in the ERA5 dataset –for 10 boosting rounds. The XGBoost model's hyperparameters were primarily kept at their default values due to the computational constraints of training on such an extensive dataset. Specifically, we used gamma=0 (minimum loss reduction required to make a further partition on a leaf node), and both subsample and colsample_bytree were set to 1.0, meaning all data points and features were used for building each tree. The L2 regularization parameter (lambda) was set to 1.0, while L1 regularization (alpha) was kept at 0. For tree construction, we employed the "hist" tree method with a "depthwise" growth policy and 256 bins for feature discretization. The model used a single tree per iteration (num_parallel_tree=1) with squared error as the objective function. The model was trained incrementally using one month of data at a time from the ERA5 dataset, spanning the time period from January 1980 to December 2017. Data from January 2018 served as the test set to evaluate performance during training in each epoch. These parameter choices balanced model complexity with computational efficiency, as the incremental training approach already imposed significant computational demands.

An early stopping mechanism was applied during each training epoch to prevent over-fitting; the training process terminated if the error on the test set did not improve for three consecutive boosting rounds. To reduce storage requirements, data for each epoch was constructed in memory at runtime, bypassing the need for permanent storage of monthly datasets.

This design resulted in progressively longer training times as epochs progressed since each new boosting round effectively added additional estimators to the model, increasing both the training complexity and the inference computational cost. This incremental training approach was essential to handle the large volume of data and to allow periodic checkpoint saves.

### 2.5.2 Applicability to high spatial resolution data

To evaluate the applicability of the trained model to higher spatial resolutions, we used data from ERA5-Land (Muñoz-Sabater et al., 2021), a dataset that provides global coverage at approximately 9 km spatial resolution and hourly temporal resolution. Our model had no exposure to ERA5-Land data during training. To assess performance in data with a different spatial resolution compared to the training data, we utilized global data from 2018 to generate hourly THI predictions using our model trained on coarser ERA5 data. These predictions were then validated against reference ground truth THI values computed directly from ERA5-Land data, following the procedures described above.

## 3  Results

### 3.1  Model Training and Evaluation

Consistent with the methodologies put forth in the Methods section, this study included the training of three XGBoost regression models, each varying in complexity, to establish the optimal parameterisation for the prediction of hourly Temperature Humidity Index (THI) values from daily climatic input. Figure 2 illustrates the progression of two key performance indicators, MAE and MSE, throughout the training phases. These metrics were computed at the end of each training epoch, corresponding to monthly intervals spanning January 1980 to December 2017. The evaluation was carried out using a test dataset, which included global data from January 2018, to validate the predictive accuracy and generalization capability.

Furthermore, the right panel of Figure 2 presents a ground truth versus prediction plot for Model 1's inference on the validation set (February 2018 - December 2020). We employed a density plot instead of a scatter plot to facilitate visualisation of the clustering behavior within the large dataset (approximately 10 billion data points). As evident, the model demonstrates good performance, with the majority of points concentrated near the diagonal, representing optimal prediction. However, a potential limitation is observed at the lower end of the Thermal Humidity Index (THI) range. Here, the model appears to exhibit a prediction floor around $\sim$-40 THI. It is important to note that these low THI values are of minimal interest for heat stress studies, as they primarily correspond to regions such as Antarctica, which are normally devoid of human and livestock populations.

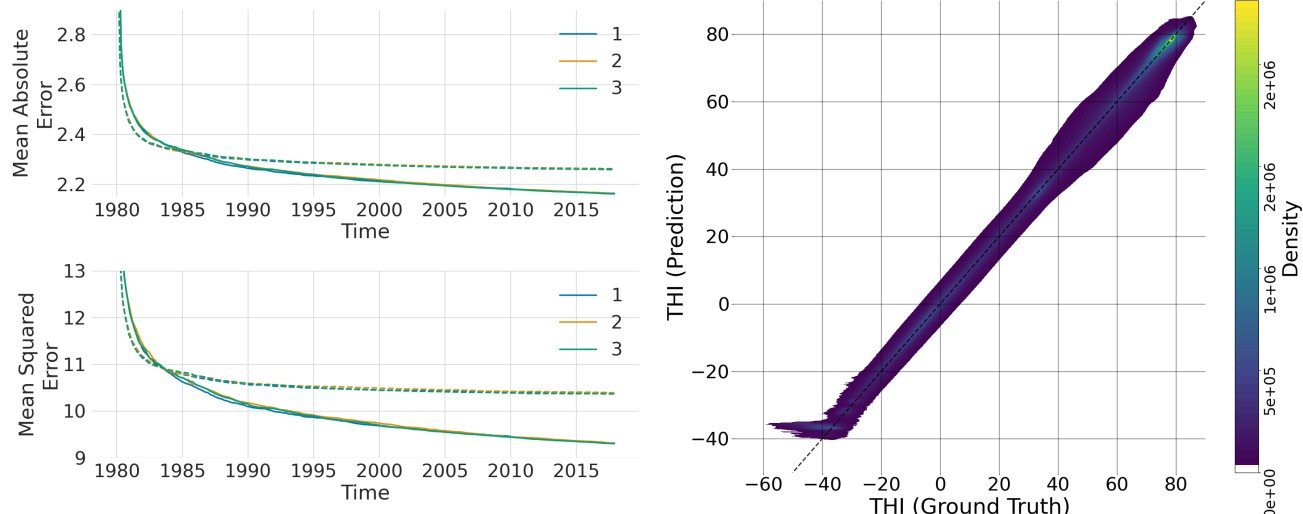

**Figure 2.** Evolution of MAE and MSE throughout the training process for three distinct XGBoost models, each represented by a different color. The left panel shows the MAE (top) and MSE (bottom) metrics over training epochs, conducted on a monthly basis from January 1980 to December 2017. Solid lines depict the metrics evaluated on the training set in each epoch, while dashed lines represent the MAE and MSE evaluated on the test set at each epoch. The right panel displays a density plot of the predictions from Model 1 versus the ground truth for the validation data set (2018-2020), where the diagonal line indicates the optimal prediction performance.

Following the approach outlined in the Methods section, the training process for the models was executed incrementally, with the dataset being segmented into monthly intervals. This approach facilitated the storage of checkpoints at the conclusion of each epoch, allowing for a systematic evaluation and resumption of the training process without loss of progress.

The performance metrics of the three models were found to be closely comparable across the evaluation criteria. Both the
250 MAE and MSE demonstrated a continuous decrease throughout the training epochs, albeit at a diminished rate of reduction as the training progressed. It is noteworthy that these metrics were also assessed using a test set that was not seen by the model during the training phase, ensuring the evaluation of the model's predictive capability on unseen data. As the complexity of the models increased, notably, Model 3 required a substantially longer duration for training compared to its counterparts. Furthermore, an observable convergence between the curves representing MSE and MAE was observed, in both the training
and test sets, indicating a stabilization in the models' performance over time. Table 4 shows the performance metrics of the three models across the training and test sets, as well as the total training time and time needed to perform inference on a single year and scenario. Lastly, the metrics obtained from the validation set were closely comparable across all three models, with MAE reaching $\sim$3.4, MSE $\sim$19, and $R^2$ $\sim$0.94.

To further assess the comparative performance of the three trained models, we performed inference using data from 2018
to 2020 (ERA5 reanalysis) to evaluate the precision of the THI predictions relative to the ground truth. Figure 3 displays THI predictions at six randomly selected grid points and time intervals, representing diverse climatic conditions: permanent

**Table 4.** MAE, MSE and $R^2$ performance metrics evaluated on the last epoch during the training process, the test set for the three models (data from Jan 2018) and the evaluation set (Feb 2018 - Dec 2020). Furthermore, the total training time and the time needed by the models to perform inference on a single year are presented.

| Model no. | Training set | | | Test set | | | Validation set | | | Total training time (hrs) | Time to evaluate 1 year (min) |
|---|---|---|---|---|---|---|---|---|---|---|---|
| | MAE | MSE | $R^2$ | MAE | MSE | $R^2$ | MAE | MSE | $R^2$ | | |
| 1 | 2.163 | 9.306 | 0.942 | 2.262 | 10.386 | 0.940 | 3.432 | 19.014 | 0.943 | **564** | $\sim$ **205** |
| 2 | 2.160 | 9.294 | 0.940 | 2.260 | 10.392 | 0.941 | **3.402** | **18.624** | **0.944** | 792 | $\sim$ 375 |
| 3 | **2.159** | **9.279** | **0.939** | **2.259** | **10.366** | **0.944** | 3.403 | 18.754 | **0.944** | 1156 | $\sim$ 480 |

frost regions from Antarctica (top row), moderate climate (middle row), and two hot climate regions (bottom row). In the two examples from Antarctica, the model was found to have a lower limit in its prediction window close to -40 THI units. Across the rest of the examples, the outputs from all three models closely followed the real THI fluctuations during the 10-day periods shown. In the examples of THI originating from colder regions of the world (middle row of panels), the THI prediction captures the average trend well, but the finer scale fluctuations are less well-represented. Additionally, the predictions generated by the three models were nearly identical, as shown in Figure 3.

The outputs from the three models on ERA5 data align well with the ground truth THI, especially in mild and hot environments. To evaluate the similarity of their performance on CMIP6 future projection data, we conducted inference using a single year of data from the ACCESS-ESM1-5 model (year 2020 under scenario SSP2-4.5) for all three models. Figure 4 presents the THI outputs at four randomly selected geographical locations and time points over a 10-day period. The outputs from all models closely match each other, corroborating their consistency. Combined with the previously obtained performance metrics, this indicates that the three models exhibit similar performance on both ERA5 and CMIP6 data. Consequently, we opted for the simplest model (Model 1) due to its significant computational cost savings compared to the other models. The marginal improvements in performance metrics did not justify the additional tens of thousands of CPU-hours required for the more complex models, given the close similarity in their outputs.

To assess model performance, we employed global maps at randomly chosen time points from the validation set. These maps show the Temperature Humidity Index (THI) using both ground-truth data and the chosen model's predictions, along with their difference. Representative examples are shown in Figure 5. Deviations from the ground truth are evident in various regions across the globe at these hourly time points.

In addition, to further assess the performance of the trained model on a spatial level, we constructed maps of ME, MAE and MSE, using the evaluation set put aside during training. These maps are presented in Fig. 6. The ME metric allowed us to quantify whether there was any systematic overestimation or underestimation in specific areas of the world. As observed, there is significant overestimation of THI in a large portion of Antarctica. This overestimation is not of concern for the scope of this study, as Antarctica is an region with no risk of heat stress for humans or livestock. Due to the low relevance of heat stress in this uninhabited region and to optimize computational resources, Antarctica was excluded from further inference procedures. This was not found in the North Pole regions, as it was excluded from training and evaluation procedures due to the absence of

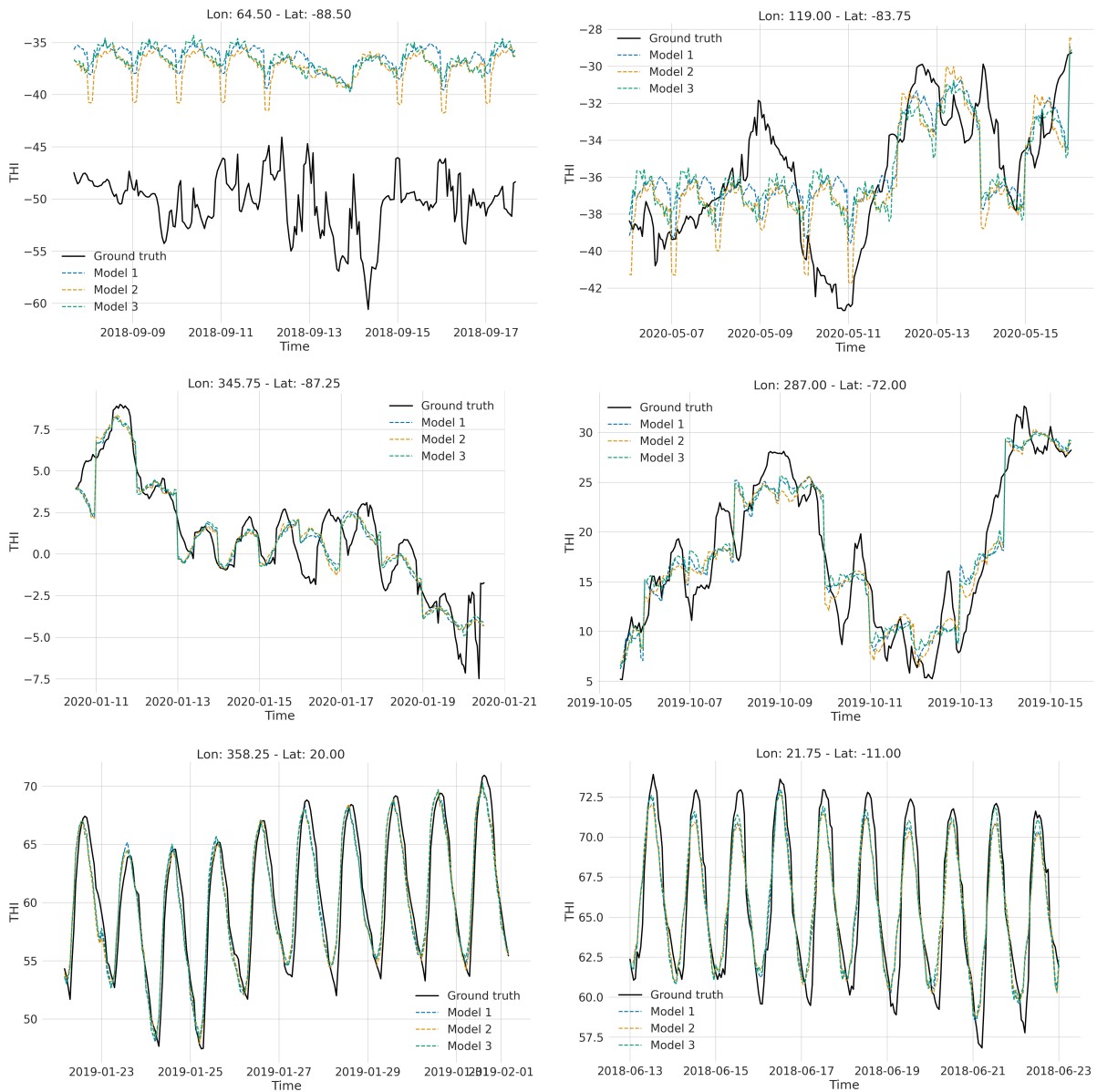

**Figure 3.** Comparative visualization of hourly THI calculations using ERA5 reanalysis data (ground truth) and the outputs from the three XGBoost models, each differentiated by unique colors. The six-panel display (3x2 arrangement) showcases THI profiles across varying climatic conditions: the top row of panels presents examples from the South Pole, where a prediction minimum of ∼-40 THI units was found in days with 0 hours of sunshine, the middle row illustrates moderate climate conditions, whereas on the bottom row of panels depicts examples from warm climate regions. This arrangement provides a comprehensive overview of the models' performance and accuracy in replicating ground truth THI values across a spectrum of environmental conditions. This comparison explains further the density q-q plot.

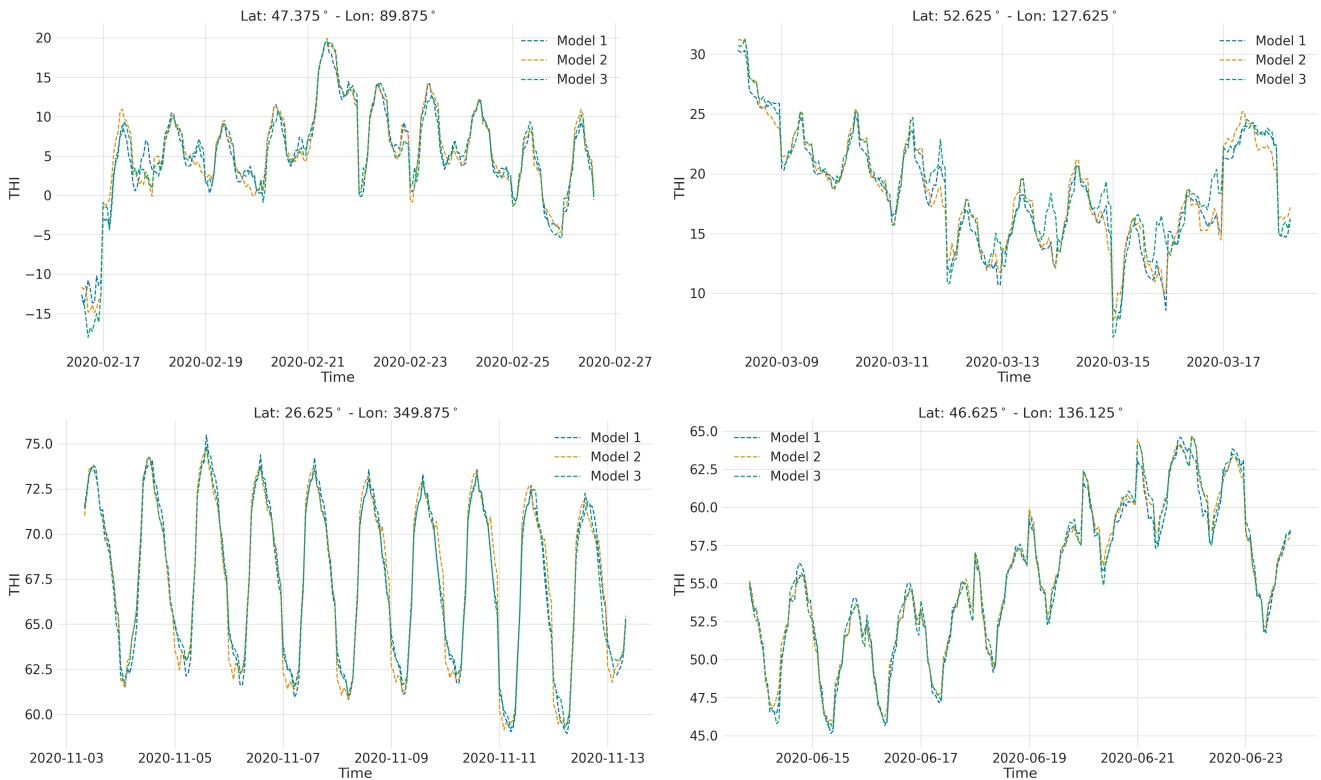

**Figure 4.** Comparative visualisation of the THI profiles from the three models' predictions from the ACCESS-ESM1-5 model (SSP2-45 scenario) at the year 2020. The four panels show four randomly selected grid points and the prediction from each model is colour-coded.

land at latitudes higher than 83° N. We attribute this overestimation to the length of the day feature, as this was only observed in months were there was no sunlight in the specific region. A month-by-month figure of ME is presented in Appendix C.

The spatial distribution of Mean Absolute Error (MAE) indicates that the model performs well in equatorial regions, accurately predicting hourly THI values with MAEs around 1 THI unit. However, MAE values are higher in some mountainous regions, such as the western United States, Tibet, and Mongolia, where they range from 4 to 6 THI units. This discrepancy is further highlighted in the Mean Squared Error (MSE), which reaches values between 25 and 35 in these areas. Since the model shows minimal Mean Error (ME) in these regions, it effectively captures the average THI conditions but struggles with the

larger diurnal temperature variations typical of high altitudes (Pepin and Seidel, 2005).

  This altitude dependent performance if clearly demonstrated in the kernel density estimation plot, presented in the bottom panel of Fig. 6, which shows a strong relationship between MAE and elevation. The KDE plot shows that the vast majority of predictions at low altitudes (below 500m) cluster around the model's mean MAE of around 2-3 THI units, an indication of consistent performance in lowland areas. There is however a positive relationship between MAE and elevation, evident by the

positive gradient of the linear regression fit. While the model may have slight inaccuracies in capturing THI at specific hours,

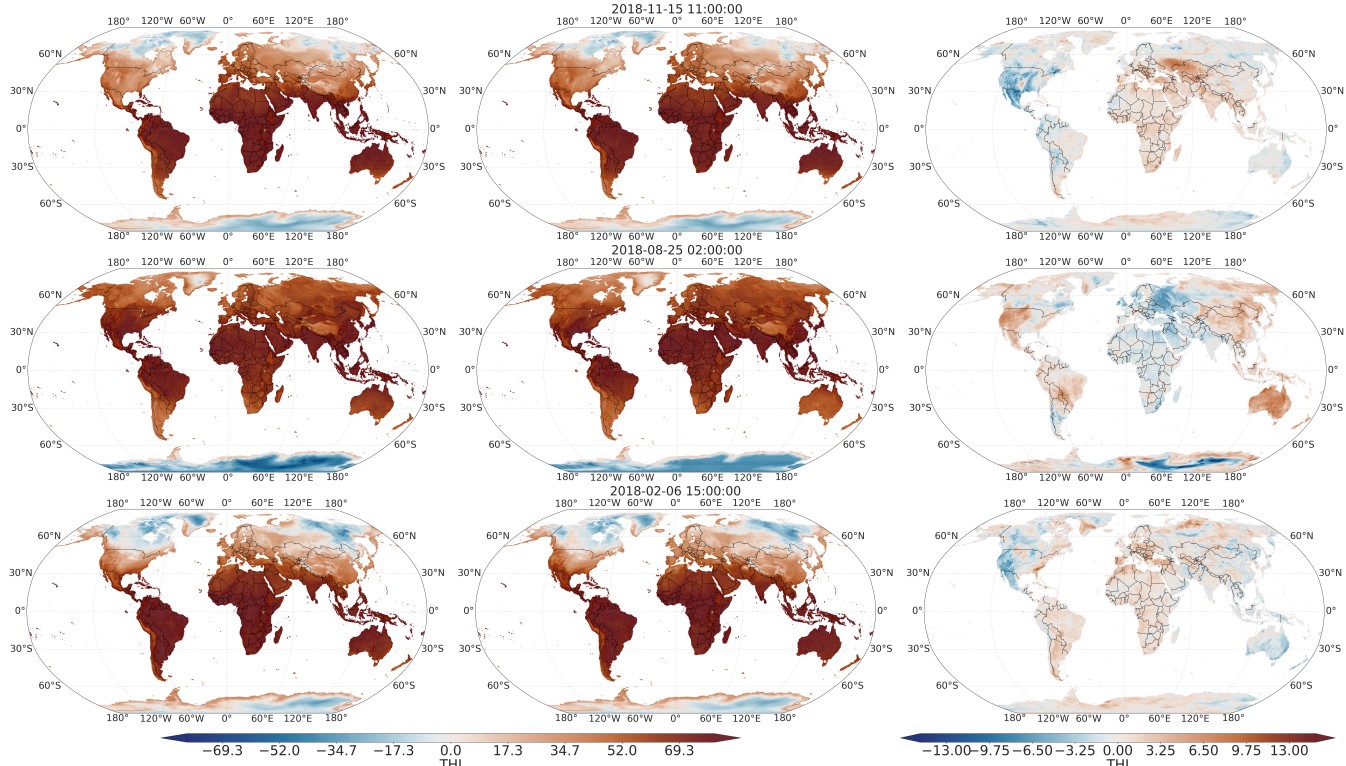

**Figure 5.** Comparison of Ground Truth THI and Model Predictions for three randomly selected hourly time-points. The first column displays the Ground Truth THI values calculated from ERA5 data for three randomly selected time points within the evaluation period. The second column shows the corresponding THI predictions from our model. The third column illustrates the differences between the Ground Truth and the model predictions (Ground Truth - Prediction).

there is no indication of a systematic bias across the dataset. These findings suggest that, overall, the model's hourly predictions are robust.

Lastly, we explored the applicability of the trained model on a different dataset with similar hourly temporal resolution but higher spatial resolution, namely ERA5-Land, which is available at a 9 km resolution (Muñoz-Sabater et al., 2021). When evaluating the model's performance on ERA5-Land data, we calculated the Mean Absolute Error (MAE), Mean Squared Error (MSE), and $R^2$ values by comparing the model's predictions against THI values derived directly from the ERA5-Land dataset (shown in Appendix B). These performance metrics were found to be closely aligned with those obtained on the original ERA5 data, demonstrating the model's consistency when applied to datasets with finer spatial resolution.

Notably, the model exhibited a similar overestimation of THI values in Antarctica on ERA5-Land as it did on ERA5. Furthermore, the spatial distribution of errors remained consistent with that observed on the ERA5 data. Detailed results of this analysis are presented in Appendix B.

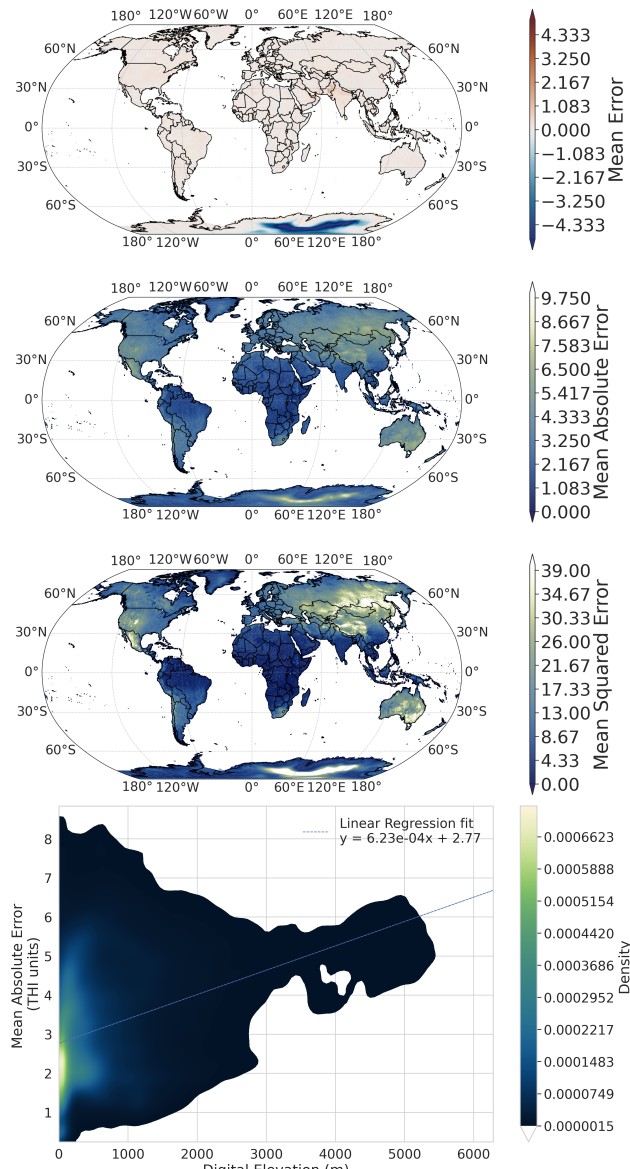

**Figure 6.** Spatial and altitudinal distribution of model performance metrics obtained from the evaluation set (Feb 2018 to Dec 2020). Top panel shows the ME, an indicator of systematic bias in the THI predictions. The second panel from the top shows MAE, an indicator of the average magnitude of prediction errors, and the third panel shows MSE, which emphasizes larger prediction discrepancies. On the bottom row, the relationship between MAE and altitude is presented as a kernel density estimation plot. The dashed line represents a linear regression fit between the two.

## 3.2 THI projections

Building upon these findings, we employed Model 1 for inference using 12 GDDP NASA-NEX CMIP6 models under two distinct climate scenarios: SSP2-4.5 (representing a moderate stabilization emission scenario) and SSP5-8.5 (representing a business-as-usual scenario with rising emissions until the end of the century). The implicit assumption in this approach is that the diel cycle of the THI does not alter significantly under climate change scenarios. Utilizing all combinations of climate models and scenarios, we generated datasets spanning the period 2020 to 2100. These datasets are publicly available at https://doi.org/10.26050/WDCC/THI for further investigation and use in climate change impact studies (Georgiades, 2024).

To address the uncertainties inherent in long-term climate projections, especially those extending to the end of the century, we employed an ensemble approach that incorporates outputs from twelve climate models. Each model is based on varying assumptions, parameterizations, and computational algorithms, which result in different projections of future conditions. By including this range of models, we capture a broad spectrum of potential climate outcomes, thereby accounting for the variability and uncertainty characteristic of long-term projections. This approach allows for the construction of projection intervals that provide a probabilistic range of possible scenarios, rather than relying on a single deterministic outcome. This ensemble method, widely adopted in climate science, allows us to average or analyze the full set of outputs, offering robust estimates that reflect a range of plausible future conditions.

## 3.3 Limitations

One limitation of the model, evident from the spatial distribution of MAE and MSE, is the reduced accuracy in regions with complex topography, such as certain mountainous areas, where MAE and MSE values are higher when compared to equatorial regions, even though the average THI is captured well (ME is close to 0). This discrepancy likely originates from the unique microclimates and larger diurnal variations often observed at higher altitudes (Pepin and Seidel, 2005). This limitation may lead to over- or under-estimation of THI values in mountainous terrains, affecting the precision of heat stress predictions for livestock in these areas.

Additionally, our model relies on daily climate projections which are temporally downscaled to an hourly resolution. While effective for capturing broad diurnal trends, this approach may not fully account for short-term extreme weather events or rapidly changing temperature and humidity conditions, especially in regions prone to sudden weather shifts.

To address these limitations, more advanced machine learning techniques could be employed, including deep learning models designed to capture complex temporal and spatial dependencies, such as transformer-based models and convolutional neural networks (Vaswani et al., 2023; Ashiotis et al., 2023). These architectures are capable of modeling intricate patterns and variability within climate data, potentially improving prediction accuracy in regions with complex topography and variable climate conditions. However, implementing these models would require significantly greater computational resources.

## 4 Conclusions

Climate change, driven by anthropogenic emissions, entails a significant risk to ecosystems and societies worldwide. One of the anticipated consequences is rising global temperatures. The agricultural sector, vital for global food security and economies, is particularly vulnerable. Dairy farming, a crucial sub-sector, faces significant economic challenges due to heat stress impacting dairy cattle and associated impacts from the exposure to heat and humidity anomalies. Heat stress in dairy cows is commonly quantified using the Thermal Humidity Index (THI), a simple metric requiring only temperature and humidity data. Previous work utilized daily THI values, lacking the necessary granularity to capture the crucial intraday climatic variability for accurate heat load estimation.

To address this limitation, we trained a machine learning model (XGBoost regressor) on global hourly historical reanalysis data (ERA5) to effectively downscale daily climate variables to hourly THI values. Our models demonstrably performed well against ground truth data from an independent validation period. The implicit assumption in this approach is that the diel cycle of the THI does not alter significantly under climate change scenarios.

Leveraging the good performance and agreement between the three models, we employed the most computationally efficient model to generate global hourly THI projections until the end of the century. This involved utilizing 0.25° GDDP NASA-NEX CMIP6 data with 12 climate models and two emission scenarios (SSP2-4.5 and SSP5-8.5).

The generated hourly THI datasets hold significant potential to contribute towards the optimization of heat stress management in the dairy industry. These datasets can empower stakeholders with the ability to create highly accurate and geographically specific heat stress risk assessments. This information can then be used to develop targeted mitigation strategies, allowing farmers, agricultural communities and organizations to proactively manage heat stress and optimize animal well-being and production efficiency. Furthermore, incorporating these datasets into climate change adaptation plans allows policymakers and the dairy cattle sector to develop long-term strategies for ensuring the sustainability of the dairy industry in the face of a changing climate. Ultimately, this research paves the way for a more resilient and sustainable future for dairy farming.

## 5 Code and data availability

Code to reproduce the results presented in this paper is available in the public github repository github.com/pantelisgeor/Temperature-Humidity-Index-ML. The code provided is written in Python and the workload can be executed by running a series of bash scripts, as documented in the repository description. The code is provided under an MIT license, which allows for users to freely use and modify the code.

The data produced in this study are available at https://doi.org/10.26050/WDCC/THI (Georgiades, 2024) in NetCDF format, with an hourly temporal resolution and 0.25° spatial resolution. The datasets are published under a CC BY 4.0 license.

## Appendix A: XGBoost and Random Forest comparison

To aid our choice of predictive algorithm, we conducted a comparison between the XGBoost and Random Forest regression algorithms, utilizing their respective implementations in Python's *xgboost* and *scikit-learn* libraries. For a fair comparison, we applied identical values for parameters that were analogous between the two models:

  – **Number of estimators:** 50

  – **Maximum depth:** 20

  – **Loss function:** Root Mean Squared Error (RMSE)

  – **Features in each tree:** Square root of the number of features

  – **Number of cores for parallelization:** 128

Both models were trained using global data from January 2000 and evaluated using data from January 2018. Table A1 presents the time taken in minutes for each model to train and evaluate one month's worth of data ($\sim$290 million data points), along with the memory utilization during training. It's noted that the times reported here exclude data loading and feature construction processes and include only the training and inference procedures for the two models. Comparing these figures, it's evident that, in the current application, the XGBoost algorithm is more efficient in both compute time and memory utilization.

**Table A1.** Time taken for training and inference procedures, and memory utilization during training for the XGBoost and Random Forest models.

| Model | Training time (min) | Inference time (min) | Memory utilization (GB) |
|---|---|---|---|
| XGBoost | 3:02 | 0:11 | $\sim$40 |
| Random Forest | 8:43 | 0:52 | $\sim$130 |

Next, we compared the predictive power of the two trained models, using global data for Jan 2018. Both models were used to predict the hourly THI values for this time period and compared to ground truth (THI values calculated using the ERA5 data). Table A2 presents the MAE, MSE and $R^2$ metrics obtained from evaluating the two models' predictions for the month of Jan 2018 against the ground truth THI calculated using the ERA5 data. In all three metrics, the XGBoost model outperformed the Random Forest model.

**Table A2.** The MAE, MSE and $R^2$ metrics obtained from comparing the two model predictions for Jan 2018 against the ground truth THI, calculated using the ERA5 data.

| Model | MAE | MSE | $R^2$ |
|---|---|---|---|
| XGBoost | 3.297 | 13.61 | 0.98 |
| Random Forest | 3.494 | 15.82 | 0.96 |

Figure A1 displays a 2x2 grid of predictions generated by the XGBoost and Random Forest models against the ground truth THI values, derived from the ERA5 dataset, for four randomly selected locations. As shown, XGBoost performs slightly better compared to the Random Forest model. Finally, an important, reason behind our decision to carry out this study using the XGBoost model was the need for incremental learning, which the Random Forest implementation lacks. Given the extensive data volume and the necessity for checkpoint saves, the XGBoost algorithm was ultimately chosen.

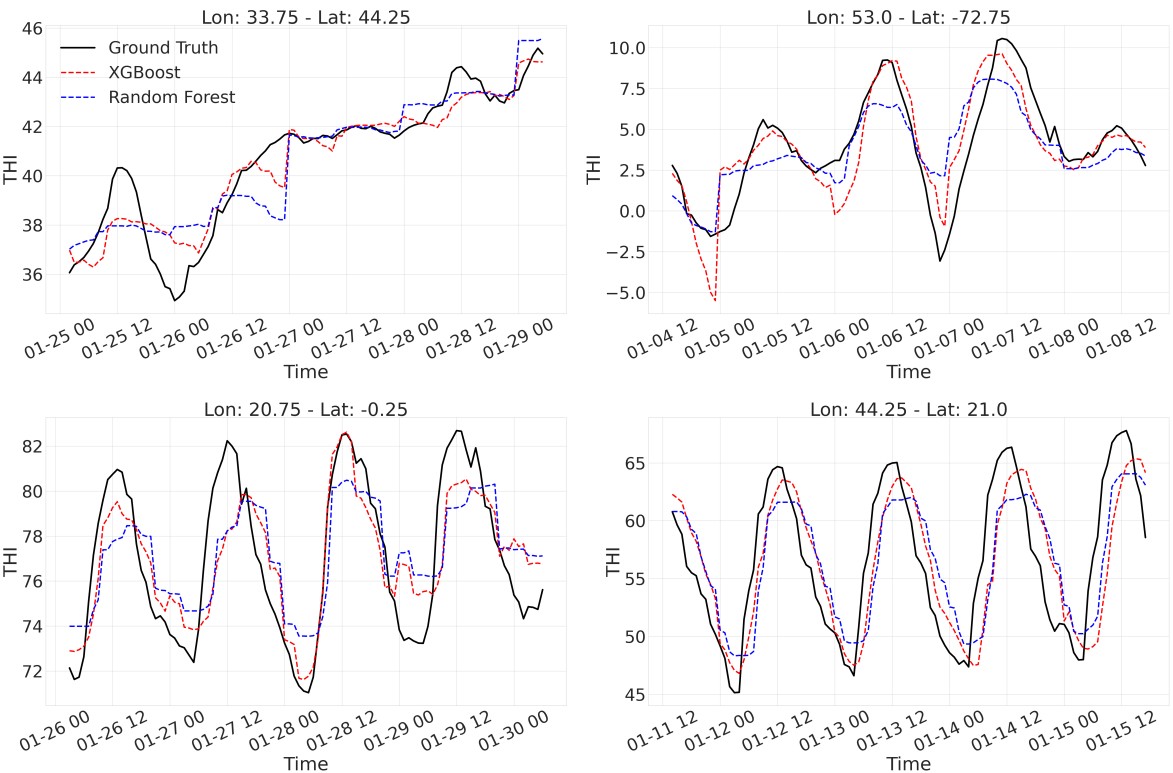

**Figure A1.** Comparative time-series plots of four randomly chosen locations between the predictions made by the XGBoost and the Random Forest against the ground truth THI values. The predictions shown here are from Jan 2018.

**Appendix B:  Applicability to high spatial-resolution data**

To evaluate the applicability of the trained model on input data with higher spatial resolution than ERA5, we utilized the ERA5-Land dataset for the year 2018. This dataset offers a spatial resolution of ~9 km and an hourly temporal resolution. The methods for feature construction employed in this experiment were identical to those used for the ERA5 dataset.

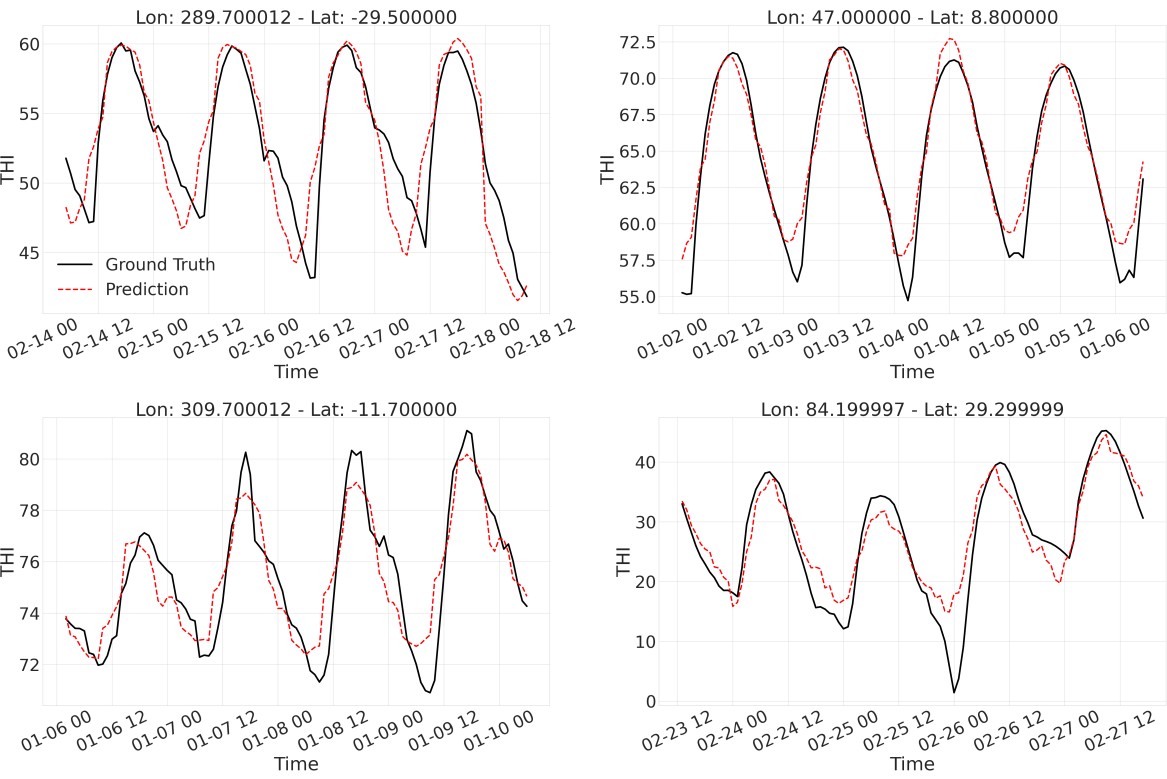

**Figure B1.** Ground truth (THI derived from ERA5-Land data) and model prediction examples for the year 2018.

Table B1 presents the evaluation metrics obtained from comparing the model predictions and THI ground truth values,
calculated using the ERA5-Land dataset for year 2018. Finally, Fig. B3 displays the spatial distribution of ME, MAE and MSE
for the model predictions against ground truth THI values, evaluated using the ERA5-Land dataset.

**Table B1.** Evaluation metrics obtained using the trained model and ERA5-Land data for the year 2018.

| MAE | MSE | $R^2$ |
|---|---|---|
| 3.941 | 19.241 | 0.957 |

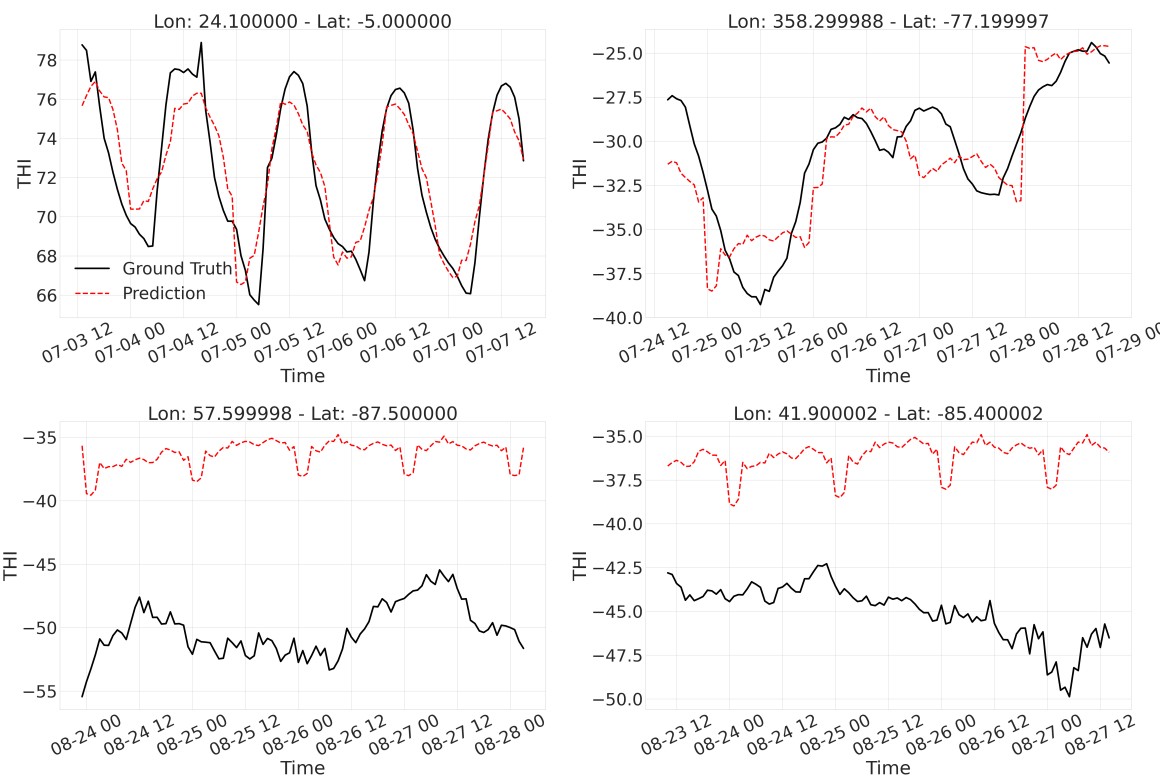

**Figure B2.** Examples of ground truth (THI derived from ERA5-Land data) and model predictions for the year 2018. The bottom panel presents cases from Antarctica, where the model the model approaches a predictive limit of -40 THI units, comparable to the inference results obtained using ERA5 data.

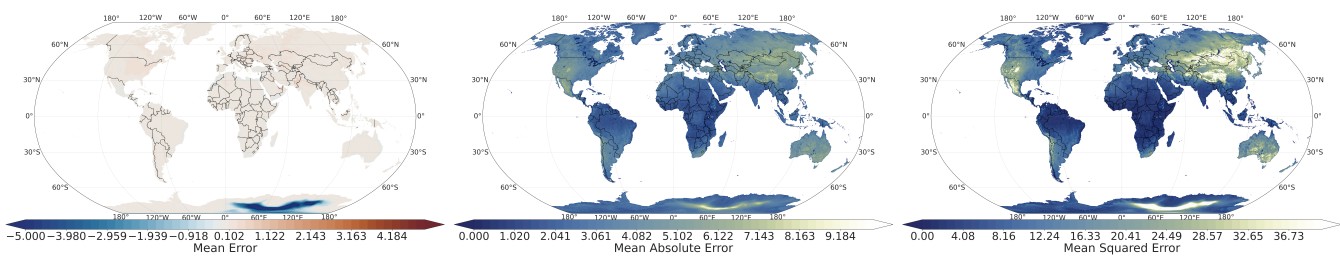

**Figure B3.** Spatial distribution of ME (left panel), MAE (middle panel) and MSE (right panel) as evaluated from ERA5-Land for the year 2018.

## Appendix C: Antarctica

We further examined the systematic overestimation of THI in Antarctica by calculating the Mean Error (ME) globally for each month of the year, as shown in Fig. C1. The results indicate that ME is minimal during months with non-zero sunlight,

while significant THI overestimation occur from March through October in Antarctica, coinciding with periods of continuous darkness at latitudes approaching -90°, as shown in Fig. C2.

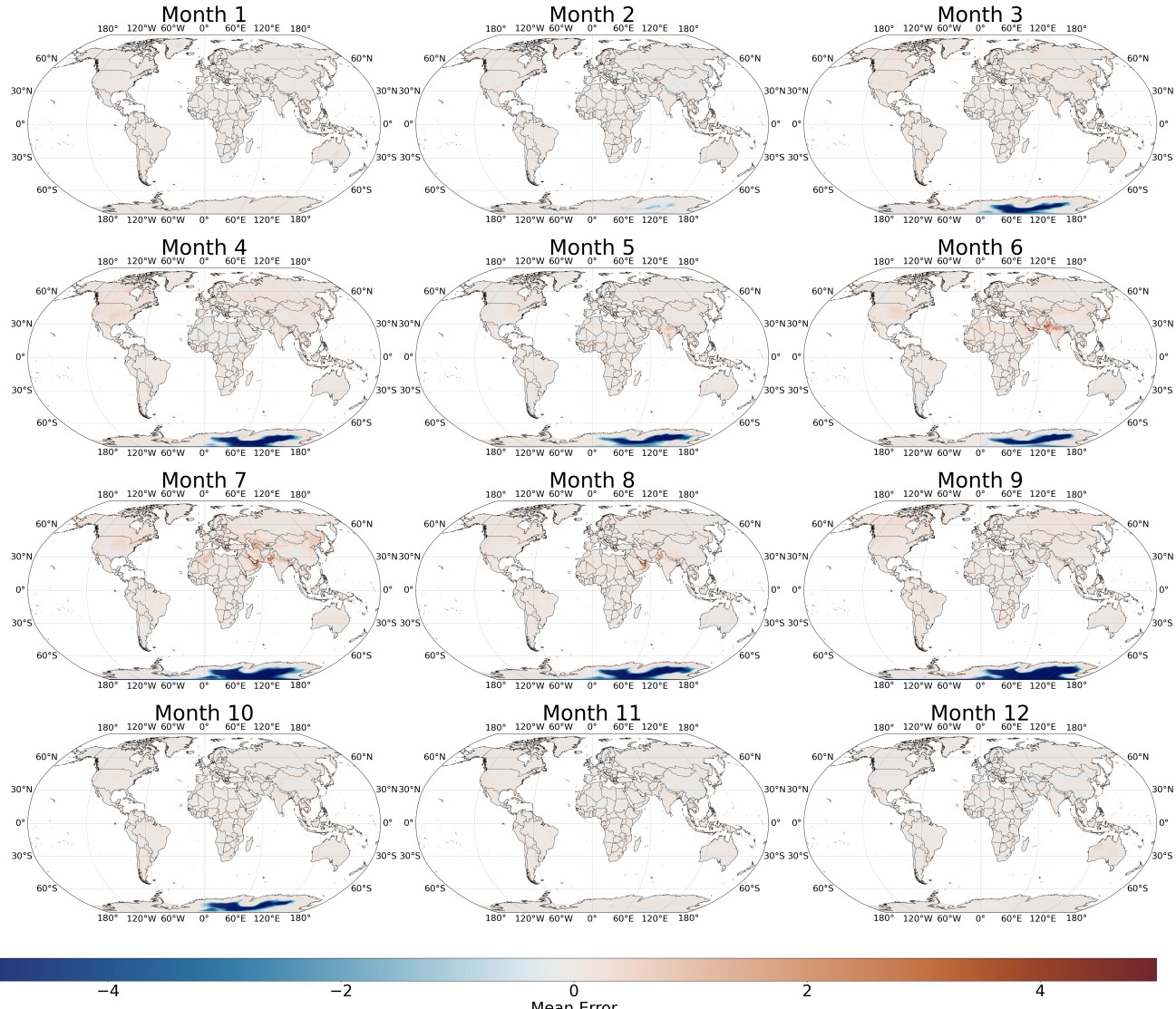

**Figure C1.** Evolution of ME in Antarctica for each month of the year between ground truth THI, calculated from ERA5, and the trained model prediction.

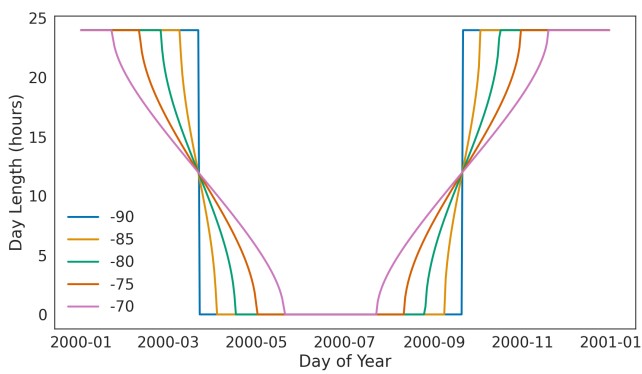

**Figure C2.** Number of hours of sunshine experienced by areas at latitudes ranging from -70° to -90° throughout a year.

*Author contributions.* PG: Conceptualization, Data curation, Format Analysis, Methodology, Software, Validation, Original draft preparation, TE: Methodology, Review & editing, YP: Data curation, Software, Resources, Review & editing, JA: Data curation, Software, Review & editing, JL: Review & Editing, MN: Conceptualization, Methodology, Validation, Original draft preparation

*Competing interests.* The authors declare that they have no conflict of interest.

*Acknowledgements.* We would like to thank the High Performance Computing Facility of the Cyprus Institute for their support in the computational and storage needs for this study.

This research was supported by the EMME-CARE project that has received funding from the European Union's Horizon 2020 Research and Innovation Program, under Grant Agreement No. 856612, as well as matching co-funding by the Government of Cyprus. A short,
comparative study between the performance and resource requirements of the XGBoost and Random Forest algorithms is presented, which supports our choice for this study. This research was also supported by the PREVENT project that has received funding from the European Union's Horizon Europe Research and Innovation Program under Grant Agreement No. 101081276 and the European High Performance Computing Joint Undertaking (JU) and Germany, Bulgaria, Austria, Croatia, Cyprus, Czech Republic, Denmark, Estonia, Finland, Greece, Hungary, Ireland, Italy, Lithuania, Latvia, Poland, Portugal, Romania, Slovenia, Spain, Sweden, France, Netherlands, Belgium, Luxembourg,
Slovakia, Norway, Türkiye, Republic of North Macedonia, Iceland, Montenegro, Serbia under grant agreement No 101101903.

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
