# Peer review of "Global Projections of Heat-Stress at High Temporal Resolution Using Machine Learning"

_Earth System Science Data, 2024_

## Referee Comment (RC2)

Earth System Science Data

General comments: The core idea and content of this manuscript focus on applying machine learning algorithms to improve the prediction accuracy of heat stress in dairy cows, especially the prediction of temperature-humidity index (THI) under the background of climate change. It enables better understanding and predictions to mitigate the impacts of climate change on livestock farming, thereby aiding in the formulation of strategies to combat climate change. However, there are some problems that have to be addressed in this manuscript.

**1. Introduction**

Specific comments:
The authors did not clearly define the differences and novelty of its research compared to existing research. Why do you use the machine learning algorithm to downscale climatic data? What is its advantage in downscaling compared to other methods? The manuscript does not show the necessity of the method.

**2. Methodology**

Specific comments:
1) There are not enough details about model training. For example, when training model, how to build one-to-one relationships between the hourly training data from ERA5 dataset and the daily inference data from both ERA5 and NEX-GDDP CMIP6 data? These descriptions are critical because they will directly decide how to realize the downscaling from daily climate variables to hourly THI values.

2) Why do you use 12 climate models? What is the meaning of selecting so many climate models?

Other comments:

(1) I suggest to provide more explanations about performance metrics such as MAE, MSE, and how they can help assess model performance.

(2) Some contents in section Results should appear in Methodology. For example, how to test and valuate the model; needed computing resources; size of trained samples.

**Results**

Specific comments:
(1) I suggest a discussion on the uncertainty of results and how to quantify this uncertainty.

(2) The manuscript does not discuss the credibility and potential limitations of long-term predictions that extend to the end of this century.

(3) I suggest clearly identify the limitations of the model, the impact of these limitations on the results, and discuss possible directions for improvement.

(4) For the results from Figure 4, the conclusion was made "Deviations from the ground truth are evident in various regions across the globe at these specific hourly time points". The analysis in Figure 4 is to further assess model performance and identify potential systematic errors. I am wondering what the potential systematic errors are from the evident deviations between the predictions and the ground truth. I suggest that the authors make some discussions about that.

(5) Authors indicated the absence of systematic errors because the mean difference between ground truth and prediction reveals minimal deviations from zero (Figure 5). This conclusion is arbitrary. It should be effect of temporal scale (grain). At finer temporal grain (such as at hourly grain), more fluctuations can be exhibited, while they become smoother at coarse temporal grain (such as at yearly grain) because delicate variations are masked.

Some descriptions are not clarified. For example,
(1) "Across these examples, the outputs from all three models closely followed the real THI fluctuations during the 10-day periods shown." This result is suitable for just two hot climate regions for the two bottom panels in Figure 2. However, in the text, it is not clarified.

(2) In the text, it is shown that "… at these specific hourly time points", however, in the caption of Figure 4, it is shown that "…for three randomly selected time points". I am wondering whether they have the same meaning.

(3) In Figure 5, it is not clear how to average the THI over the three-year period. Are all the hourly THI values (at all the time points) in the validation set (February 2018 - December 2020) were averaged?

Other comments:

It is not necessary to give the explanations in the whole names again and again after their simplified expressions have been made in the preceding text, for instance, Mean Absolute Error (MAE) and Mean Squared Error (MSE).

**Conclusion**

"The implicit assumption in this approach is that the diel cycle of the THI does not alter significantly under climate change scenarios." This assumption is not provided over the entire manuscript. Why does it appear in section Conclusion?

---

## Author Comment (AC1)

**Response to the reviewers**

The authors thank the reviewers for their time and effort in providing critical evaluation of our work. In the following, we address their concerns point by point.

**Reviewer 1**

This article employs the XGBoost model to temporally downscale daily climate data, generating THI data that quantifies the impact of temperature and humidity on cattle. The model training requires 24-48 days, but generating a year's data only takes 3-8 hours. Overall, the topic is interesting. However, I have some concerns as follows:

- 1. "We opted for the XGBoost model for its computational efficiency compared to Random Forest and other analogous algorithms, specifically for our use case." The authors claim that XGBoost offers higher computational efficiency than Random Forest and similar algorithms in their specific case. Please provide specific comparative values.
  - We have added an Appendix section (Appendix A), which presents results obtained from a comparison between the XGBoost and Random Forest algorithms. This comparison shows the higher computational efficiency and slightly improved performance of the XGBoost model relative to a Random Forest model with comparable hyperparameters.
  - We would like to note that the short inference times reported in Appendix A do not represent the inference time required by the fully trained XGBoost model, as the iterative training procedure incrementally added 10 estimators per training step, increasing both computational cost and predictive performance.
- 2. The spatial resolution of the THI is 0.25°, which seems to be a very coarse pixel size. How can you ensure that the values accurately represent such a large spatial area? Additionally, the input data for the method includes ERA5 and NEX-GDDP-CMIP6 with a spatial resolution of 0.25° as well. Would this method still be applicable if high-resolution data were available?
  - While our choice of training and inference datasets was largely influenced by the close match between their spatial resolution, we tested the trained model against a higher spatial resolution dataset, ERA5-Land. The model performed similarly well on ERA5-Land compared to ERA5, on which was trained on. We have added this comparison in Appendix B.
- 3. The ERA5 reanalysis dataset provides historical hourly data. Knowing its quality would be helpful. Can you utilize ground-truth data to validate the used variables in the ERA5 dataset? Without assurance of the input data quality, the quality of the trained model cannot be guaranteed.
  - Our study necessitated the availability of global, continuous time-series data for temperature and relative humidity, to optimally train our models. As ERA5 is currently regarded as the state-of-the-art for the observed atmospheric conditions at a global scale, we opted for its use. Furthermore, the performance of the reanalysis dataset has been extensively investigated in the scientific literature.

- We have added a paragraph in section 2.1.1 which expands on our reasoning behind using the ERA5 dataset and providing references to studies which investigate it performance and a relevant study that the dataset was employed to estimate thermal stress indices.
- 4. More details on the structure of the XGBoost regressor model are needed.
  - We have expanded Section 2.4.1 to provide further details on the implementation of the XGBoost regressor model.
- 5. The formula for THI indicates that its value depends on temperature and relative humidity, with the latter being calculable from ambient and dew point temperatures using equations 1-3. This suggests a strong dependency of THI on these two temperature data. An error analysis will be helpful. The author could do an experimental study to explore how errors in these temperatures affect THI accuracy.
  - Please see response to the subsequent comment.
- 6. "The performance of the trained models was assessed using ground truth data derived from the ERA5 dataset." Please provide more details about the ground truth data, including its spatial representativeness and temporal resolution.
  - We address the previous and the current comments (5 and 6) as follows: The Temperature Humidity Index (THI) is not a directly measured physical quantity, but rather a calculated metric derived from temperature and relative humidity. By utilizing the current state-of-the-art global climate dataset (ERA5), we are confident that our approach provides the most accurate representation of THI possible from gridded global data. We have expanded section 2.2 to explicitly clarify that THI is a calculated rather than a measured quantity and to emphasize that the spatial and temporal resolution of our THI data precisely mirrors the original ERA5 dataset resolution.

**Reviewer 2**

General comments: The core idea and content of this manuscript focus on applying machine learning algorithms to improve the prediction accuracy of heat stress in dairy cows, especially the prediction of temperature-humidity index (THI) under the background of climate change. It enables better understanding and predictions to mitigate the impacts of climate change on livestock farming, thereby aiding in the formulation of strategies to combat climate change. However, there are some problems that have to be addressed in this manuscript.

**1. Introduction**

Specific comments:

1. The authors did not clearly define the differences and novelty of its research compared to existing research. Why do you use the machine learning algorithm to downscale climatic data? What is its advantage in downscaling compared to other methods? The manuscript does not show the necessity of the method.

- We thank the reviewer for the comment. Our study focuses specifically on the temporal downscaling of THI, which is novel on its own, since no other similar study exists. As the goal of the study was to produce a multi-year, multi-model and multi-scenario dataset for accurate assessment of climate change induced heat stress, we opted for a machine learning approach, which is highly scalable. Lastly, regarding the need for temporal downscaling of THI, our analysis addresses the limitations of existing models that often rely on simplified assumptions, such as the perfect counter-cyclical relationship between temperature and humidity, as proposed by St-Pierre et al. (2003). These models typically provide only approximate estimations and fail to account for the real-world complexities of climatic cycles influenced by geographic diversity and seasonality.
- We have included two paragraphs in the introduction section which discuss the aforementioned points.

**2. Methodology**

Specific comments:

- 1. There are not enough details about model training. For example, when training model, how to build one-to-one relationships between the hourly training data from ERA5 dataset and the daily inference data from both ERA5 and NEX-GDDP CMIP6 data? These descriptions are critical because they will directly decide how to realize the downscaling from daily climate variables to hourly THI values.
  - We appreciate the reviewer's comments regarding the need for more details about the model training process. To clarify, we have constructed daily-level features from the ERA5 dataset to establish a one-to-one relationship between the hourly training data from ERA5 and the daily inference data from both the ERA5 and NEX-GDDP CMIP6 datasets. This approach ensures that the model can effectively downscale daily climate variables to hourly THI values.
  - We have updated the relevant section in the Methods to provide a clearer explanation of our methodology. Additionally, we have included a new subsection (2.3 Data workflow), which outlines the data workflow in detail.
- 2. Why do you use 12 climate models? What is the meaning of selecting so many climate models?
  - The selection of 12 climate models was based on our aim to account for the diversity of projections, as different climate models often produce varying projections due to differences in parameterization, assumptions etc. By utilizing a diverse set of models, we aim to capture a broader range of potential future scenarios. Additionally, this aids in mitigating the uncertainty associated with using a single model prediction. By averaging the outputs or analyzing the ensemble, we can achieve more reliable estimates that reflect a range of possible climatic conditions, which is aligned with established practices in climate science (Fischer et. al 2015). For example, the IPCC and other assessments commonly apply ensemble climate modeling to evaluate climate change impacts and support the development of adaptation strategies (IPCC, 2022).

- Fischer, E. M., & Knutti, R. (2015). Anthropogenic contribution to global occurrence of heavy-precipitation and high-temperature extremes. Nature Climate Change, 5(6), 560–564. doi:10.1038/nclimate2617

- IPCC, 2022: Climate Change 2022: Impacts, Adaptation, and Vulnerability. Contribution of Working Group II to the Sixth Assessment Report of the Intergovernmental Panel on Climate Change [H.-O. Pörtner, D.C. Roberts, M. Tignor, E.S. Poloczanska, K. Mintenbeck, A. Alegría, M. Craig, S. Langsdorf, S. Löschke, V. Möller, A. Okem, B. Rama (eds.)]. Cambridge University Press. Cambridge University Press, Cambridge, UK and New York, NY, USA, 3056 pp., doi:10.1017/9781009325844

**Other comments:**

- 1. I suggest to provide more explanations about performance metrics such as MAE, MSE, and how they can help assess model performance.
  - We replaced Figure 5 with the spatial distribution of Mean Error, Mean Absolute Error and Mean Squared Error, instead of the mean difference between prediction and ground truth. We have also added two paragraphs discussing these metrics and how they can be interpreted to assess the performance of the model.
- 2. Some contents in section Results should appear in Methodology. For example, how to test and valuate the model; needed computing resources; size of trained samples.
  - We thank the reviewer for this suggestion. We have moved all relevant content to the Methods section.

**Results**

Specific comments:

- 1. I suggest a discussion on the uncertainty of results and how to quantify this uncertainty.
  - We have discussed the uncertainty of our results further in the Results section (last two paragraphs).
- 2. The manuscript does not discuss the credibility and potential limitations of long-term predictions that extend to the end of this century.
  - We acknowledge the limitation of using long-term climate projections and the inherent uncertainties that are associated with it. This was the main driver of including 12 different climate models in the inference phase of the study, as it allows for the estimation of projection intervals and provides a range of plausible future climate conditions. We have added a paragraph at the end of the Results section that discusses the potential limitation.
- 3. I suggest clearly identify the limitations of the model, the impact of these limitations on the results, and discuss possible directions for improvement.
  - We have added a limitations sub-section, which elaborates on this.

- 4. For the results from Figure 4, the conclusion was made "Deviations from the ground truthare evident in various regions across the globe at these specific hourly time points". The analysis in Figure 4 is to further assess model performance and identify potential systematic errors. I am wondering what the potential systematic errors are from the evident deviations between the predictions and the ground truth. I suggest that the authors make some discussions about that.
  - The results in Figure 4 display hourly data, highlighting specific instances where the model predictions deviate from the ground truth THI values across various regions. These deviations suggest that while the model captures the general trends in THI, it may not fully capture the finer variations, resulting in differences of a few THI units compared to the ground truth. However, as shown in Figure 5, these instantaneous deviations on an hourly scale do not result in any consistent over- or under-estimation of THI. This is evidenced by the Mean Error (ME) values being close to zero across most regions, apart from Antarctica, where other factors (day length) contribute to larger discrepancies. Thus, while the model may have slight inaccuracies in capturing THI at specific hours, there is no indication of a systematic bias across the dataset. These findings suggest that, overall, the model's hourly predictions are balanced. We have included these points in our discussion of the results.
- 5. Authors indicated the absence of systematic errors because the mean difference between ground truth and prediction reveals minimal deviations from zero (Figure 5). This conclusion is arbitrary. It should be effect of temporal scale (grain). At finer temporal grain (such as at hourly grain), more fluctuations can be exhibited, while they become smoother at coarse temporal grain (such as at yearly grain) because delicate variations are masked. Some descriptions are not clarified. For example,

(1) "Across these examples, the outputs from all three models closely followed the real THI fluctuations during the 10-day periods shown." This result is suitable for just two hot climate regions for the two bottom panels in Figure 2. However, in the text, it is not clarified.

• We have changed the wording in the text to clarify this in the text of the manuscript.

(2) In the text, it is shown that "... at these specific hourly time points", however, in the caption of Figure 4, it is shown that "... for three randomly selected time points". I am wondering whether they have the same meaning.

• We have clarified in the caption that these represent three randomly selected time-points.

(3) In Figure 5, it is not clear how to average the THI over the three-year period. Are all the hourly THI values (at all the time points) in the validation set (February 2018 - December 2020) were averaged?

• We have replaced Figure 5 (now Figure 6) with the spatial distribution of ME, MAE and MSE. In essence, ME shows the same average error as before, where we averaged over all the validation set (Feb 2018 - Dec 2020), to assess whether the model was consistently over- or under-estimating THI at any region of the world.

**Other comments:**

- 1. It is not necessary to give the explanations in the whole names again and again after their simplified expressions have been made in the preceding text, for instance, Mean Absolute Error (MAE) and Mean Squared Error (MSE).
  - We have revised the manuscript to include only the acronyms of Mean Absolute Error and Mean Squared Error after the first definition of the metrics.

**Conclusion**

- 1. "The implicit assumption in this approach is that the diel cycle of the THI does not alter significantly under climate change scenarios." This assumption is not provided over the entire manuscript. Why does it appear in section Conclusion?
  - We have included this statement earlier in the manuscript.

---

## Referee Report (RR1)

Referee's report for "Global Projections of Heat-Stress at High Temporal Resolution Using Machine Learning"

The authors of this paper applied machine learning to provide temporally downscaled data and outputs of the temperature-humidity index (THI) for the cattle sector. The manuscript has undergone significant modification since the first round of review, and the authors have responded thoroughly to the suggestions made. I have two main suggestions to make that I believe would lead to a publishable article.

Firstly, the authors could provide more detail on the THI. It is difficult to interpret the variations and errors presented and discussed throughout the paper without greater context. The authors refer to specific THI thresholds but do not describe them nor their methods' impact on them, potentially missing a way of demonstrating the added value of temporally downscaling the THI.

Secondly, the authors could provide a plot relating mean absolute error to altitude. The dataset has spatially dependent errors, attributed by the authors primarily to altitude: "This discrepancy likely originates from the unique microclimates and larger diurnal variations often observed at higher altitudes". It would therefore be sensible to plot MAE against altitude to explore this attribution. This could be included as another panel in Figure 6.

I also have two minor suggestions. The first is to avoid using the blue-to-neutral-to-red colourmap for data that doesn't centre around zero. The second is to expand Figure C1 to include the entire globe, rather than just Antarctica, as it would be interesting to explore the seasonal variation of MAE across the entire globe.

---

## Author Response (AR2)

**Response to the reviewers**

The authors thank the reviewers for their time and effort in providing critical evaluation of our work. In the following, we address their comments point by point.
* * *
**Reviewer 1**

The authors of this paper applied machine learning to provide temporally downscaled data and outputs of the temperature-humidity index (THI) for the cattle sector. The manuscript has undergone significant modification since the first round of review, and the authors have responded thoroughly to the suggestions made. I have two main suggestions to make that I believe would lead to a publishable article.

Firstly, the authors could provide more detail on the THI. It is difficult to interpret the variations and errors presented and discussed throughout the paper without greater context. The authors refer to specific THI thresholds but do not describe them nor their methods' impact on them, potentially missing a way of demonstrating the added value of temporally downscaling the THI.

⇒ We have expanded the Introduction section (lines 43-50) to provide specific THI thresholds which have been shown in the literature to correspond to detrimental physiological responses in dairy cattle.

Secondly, the authors could provide a plot relating mean absolute error to altitude. The dataset has spatially dependent errors, attributed by the authors primarily to altitude: "This discrepancy likely originates from the unique microclimates and larger diurnal variations often observed at higher altitudes". It would therefore be sensible to plot MAE against altitude to explore this attribution. This could be included as another panel in Figure 6.

⇒ We have expanded Figure 6 to include a KDE plot of the MAE against the elevation, as obtained from the evaluation set. We have also included a linear regression fit to this data, which clearly indicates a positive relationship between MAE and altitude. We have also expanded our discussion to include the new panel in Figure 6 in the main body of the manuscript.

I also have two minor suggestions. The first is to avoid using the blue-to-neutral-to-red colourmap for data that doesn't centre around zero. The second is to expand Figure C1 to include the entire globe, rather than just Antarctica, as it would be interesting to explore the seasonal variation of MAE across the entire globe.

⇒ We have changed the colormap of Figures 6 and B3 to avoid using a diverging colormap on plots that don't center around 0. We have also expanded Figure C1 to include the whole globe instead of only Antarctica and expanded the text in Appendix C accordingly.
* * *
**Reviewer 2**

In this revision, most of my major concerns have been addressed. However, for response 4, the section number seems to be incorrect, as Section 2.4.1 is not present in the manuscript. Additionally, the structure of the XGBoost regressor model is still incomplete. Providing more details on hyperparameter tuning (e.g., gamma, subsample, colsample_bytree), feature interactions, and other model-specific adjustments would be beneficial.

⇒ We thank the reviewer for spotting this mistake and apologise for the wrong reference, we were referring to Section 2.5.1 (Implementation Details). Furthermore, we have further expanded the section to include all the details of the XGBoost model used in the study (lines 207-216).